# Histopathological features of prostate cancer conspicuity on multiparametric MRI: protocol for a systematic review and meta-analysis

Joseph M Norris [ORCID],[1] Lina M Carmona Echeverria,[1] Benjamin S Simpson [ORCID],[1] Rhys Ball,[2] Alex Freeman,[2] Daniel Kelly [ORCID],[3] Alex Kirkham,[4] Hayley C Whitaker,[1] Mark Emberton[1]

JMN and LMCE are joint first authors.

For numbered affiliations see end of article.

**Correspondence to**
Joseph M Norris;
joseph.norris@ucl.ac.uk

## ABSTRACT

**Introduction** Multiparametric MRI (mpMRI) has improved risk stratification for men with suspected prostate cancer. Indeed, mpMRI-visible tumours tend to be larger and of higher pathological grade than mpMRI-invisible tumours; however, concern remains around significant cancer that is undetected by mpMRI. There has been considerable recent interest to investigate whether tumour conspicuity on mpMRI is associated with additional histopathological features (including cellular density, microvessel density and unusual prostate cancer subtypes), which may have important clinical implications in both diagnosis and prognosis. Furthermore, analysis of these features may help reveal the radiobiology that underpins the actual mechanisms of mpMRI visibility (and invisibility) of prostate tumours. Here, we describe a protocol for a systematic review of the histopathological basis of prostate cancer conspicuity on mpMRI.

**Methods and analysis** A systematic search of the MEDLINE, PubMed, Embase and Cochrane databases will be conducted. The Preferred Reporting Items for Systematic Reviews and Meta-Analyses (PRISMA) guidelines will be used to guide screening, thematic reporting and conclusions drawn from all eligible studies. Included papers will be full-text, English-language articles, comparing the histopathological characteristics of mpMRI-visible lesions and mpMRI-invisible tumours. All studies published between January 1950 and January 2020 will be eligible for inclusion. Studies using confirmatory immunohistochemistry for the identification of immune subsets or structural components will be included. Study bias and quality will be assessed using a modified Newcastle-Ottawa scale. To ensure methodological rigour, this protocol is written in accordance with the PRISMA Protocol 2015 checklist. If appropriate, a meta-analysis will be conducted comparing histopathological feature frequency between mpMRI-visible and mpMRI-invisible disease.

**Ethics and dissemination** No ethical approval will be required as this is an academic review of published literature. Findings will be disseminated through publications in peer-reviewed journals and presentations at national and international conferences.

**PROSPERO registration number** CRD42020176049

## Strengths and limitations of this study

► This work will represent the first detailed systematic review and meta-analysis of the histopathological features of multiparametric MRI (mpMRI)-visible and mpMRI-invisible prostate cancers following the methodological steps of the Preferred Reporting Items for Systematic Reviews and Meta-Analyses (PRISMA) guidelines.

► The level of heterogeneity found between the reviewed articles may somewhat limit the generalisability of derived results.

► The evidence surrounding important histopathological features of mpMRI-visible and mpMRI-invisible tumours is growing; however, as this is still a new area of research, the strength of the conclusions may be limited by finite extant literature.

## BACKGROUND

The introduction of multiparametric MRI (mpMRI) has greatly improved the approach to prostate cancer diagnosis, offering improved prebiopsy risk stratification.[1] Accurate identification of high-risk disease, before a biopsy is conducted, enables increased detection of significant cancer and reduced detection of insignificant cancer, compared with traditional diagnostic approaches.[2] However, approximately 10%–20% of clinically significant disease may be overlooked by mpMRI.[1 3] The mechanisms that underpin mpMRI invisibility and the clinical implications that arise from this phenomenon have been the subject of intense research in recent years.

Prostate cancer visibility on mpMRI is positively associated with both tumour size and Gleason grade,[1 4] which suggests tumour conspicuity can provide useful prognostication, as these pathological features at biopsy are typically considered to have the strongest

**BMJ**

impact on clinical outcome. Furthermore, it appears that this is corroborated at the molecular level, as mpMRI-visible tumours are enriched with genomic features of disease aggressivity.[5 6] More recently, nuanced microstructural and pathological features have also been linked with disease conspicuity, including cellular and microvessel density, cribriform and intraductal cancer patterns, and stromal and luminal (to malignant cell) ratios.[7–11] Taken together, this evidence suggests that the visibility status of a tumour on mpMRI may be an indicator of clinical risk. Early evidence has shown that low-risk Gleason grade group (GGG) 1 tumours with a positive mpMRI at baseline are at increased risk of intervention, upgrading and unfavourable disease at the time of radical prostatectomy, compared with those with GGG1 mpMRI-invisible tumours.[12] However, mpMRI-targeted biopsy may not be insufficient alone to estimate risk of disease progression.[13] As the ability of mpMRI to detect tumours is influenced by additional histopathological features, it is pertinent to now draw together this expanding radiopathological literature to address this important clinical challenge.

The aim of this systematic review was to appraise and collate, for the first time, the evidence supporting the histopathological basis of tumour visibility and invisibility on mpMRI, in order to reveal the potential mechanisms that underpin mpMRI conspicuity, and the possible diagnostic and prognostic implication of mpMRI phenotypes.

## METHODS AND ANALYSIS

This systematic review protocol has been written in line with the Preferred Reporting Items for Systematic Reviews and Meta-Analyses Protocol (PRISMA) 2015 checklist.[14] Once identified, included studies will undergo analysis and thematic synthesis to derive the key histopathological elements associated with tumour visibility and invisibility on mpMRI.

### Search methodology

The MEDLINE, PubMed, Embase and Cochrane databases will be systematically searched in order to retrieve all studies that contribute relevant evidence. The search strategy will include Medical Subject Headings (MeSH) terms, as well as free text, joined with appropriate Boolean operators. The search will include the terms 'prostate,' 'cancer' and 'MRI', as well as multiple synonyms for the terms 'pathology' and 'histology'. The selected search terms are broad, with high sensitivity for identification of relevant studies. To include the maximum number of articles, all studies published between January 1950 and January 2020 will be eligible. We will expedite the systematic review process by uploading all articles to Rayyan, a semiautomated tool designed to improve the speed and reporting accuracy during the initial screening process and to allow three reviewers to filter duplicate studies and screen articles for relevance.[15] To further improve the evidence yield, all included articles will be reference-searched manually to identify missed studies or additional

data. Finally, experts will be consulted to identify additional literature. In the case of missing or unclear data, the corresponding authors will be contacted directly.

### Study selection and data extraction

Three researchers will independently screen eligible studies, removing irrelevant studies based on titles and abstracts. Those studies which pass the initial screen will be downloaded and the full-text examined to confirm eligibility. Disagreement between reviewers will be discussed until a consensus is reached or a fourth reviewer will be consulted. All exclusions will be noted for later analysis and the reasons for exclusion documented in detail in order to generate the PRISMA flow diagram.

### Inclusion criteria

To be included in the analysis, studies must investigate one or more histopathological aspects of the appearance of prostate cancer on mpMRI. Investigations may be conducted at the macroscopic (eg, tumour size) and microscopic (eg, morphological and pathological patterns) levels; or be based on routine or special staining, including immunohistochemistry (eg, confirmation of microstructural components, such as CD31 for microvessels).

### Exclusion criteria

Non-English language articles, conference abstracts, review articles, correspondence articles, expert opinions and case reports will be excluded. Studies that do not correlate mpMRI phenotypes with histopathological features will be excluded. Articles focusing solely on molecular characteristics (genetic or transcriptomic) or clinical features of mpMRI conspicuity will also be removed.

### Data extraction

All relevant articles will be carefully read and themes extracted. All extracted data will be held on a shared datasheet and confirmed by at least three independent reviewers to maintain veracity. Data will be collated in a manner that we have successfully demonstrated previously.[6] Table 1 summarises data items to be collected.

### Endpoints

The primary endpoint will be statistically significant differences in quantitative measurements (eg, frequency or density) of histopathological features between mpMRI-visible and mpMRI-invisible prostate cancers. Secondary endpoints will include explanatory links between these features and mpMRI conspicuity, as well as the potential clinical implications. As study methodology may impact estimated frequencies of histopathological features, we will include these variables in a moderator analysis (see Meta-analysis section).

### Risk of bias in individual studies

A modified Newcastle-Ottawa score (originally constructed for assessment of observational cohort

**Table 1** Data collection items

| Item | Data title | Data type |
|------|-----------|-----------|
| 1 | Year of publication | Study characteristic |
| 2 | Study authors | Study characteristic |
| 3 | Experimental design | Study characteristic |
| 4 | Patient population | Demographics |
| 5 | Study size | Demographics |
| 6 | mpMRI scoring scheme used | Methodology |
| 7 | Definition for clinically significant disease | Methodology |
| 8 | Definition for lesion visibility and invisibility | Methodology |
| 9 | Sample processing approach | Methodology |
| 10 | Histopathological feature studied | Outcome |
| 11 | Differential quantification of feature | Outcome |

mpMRI, multiparametric MRI.

studies) will be used to score the bias and quality across included studies.[16] This system is divided into three core elements: selection, comparability and outcome. Within each section, there are subquestions to evaluate the quality of the research methodology, at the study level. Three reviewers will be involved with this process, and any disagreement will be settled by consensus. The outcome of the bias and quality assessment will inform the thematic synthesis by providing an assessment of the reliability and applicability of the available evidence. If studies are deemed to be of excessively low quality (or high bias), then these may be excluded or, if included, will be accompanied by appropriate commentary in the discussion and conclusion. As this review is focused on the histopathological characteristics associated with prostate cancer conspicuity on mpMRI (as opposed to treatment outcome), we will modify non-applicable sections of the scoring scheme to more closely reflect the nature of the evidence base and reduce reporting inaccuracy, in a similar approach that we have taken previously.[6]

### Meta-analysis

If there are a sufficient number of studies available (ie, over three) that analyse a particular histopathological feature using a similar methodology, then we will conduct a meta-analysis. The frequency of the histopathological feature between mpMRI-visible and mpMRI-invisible tumours will be compared. Any analysis will be performed as previously described.[17] Briefly, numbers of positive and negative cases of a particular histopathological feature would be extracted from a given study and raw/direct proportions will be calculated. The distribution of untransformed, logit and double arcsine transformed

proportions will be compared. Whichever distributions resemble a normal distribution (assessed using density plots and Shapiro-Wilk tests) will be used for further analysis.

The model fitted will be determined based on inter-study variation (measured via $I^2$); if significant, a random-effect model will be fitted, or a fixed-effect model if not. After fitting a model to all relevant studies, leave-one-out (LOO) analyses and accompanying diagnostic plots will be used to identify influential studies, including externally studentised residuals, difference in fits values, Cook's distances, covariance ratios, LOO estimates of the amount of heterogeneity, LOO values of the test statistics for heterogeneity, hat values and weights. Studies with a statistically significant influence on the fitted model will be removed as outliers and the model refitted. These outliers will be examined for potential confounding variables such as study methodology or poor interobserver agreement for mpMRI scans. Finally, the summary estimates will be compared between mpMRI-visible and mpMRI-invisible cancers as subgroups, and the significance of differences will be assessed. If appropriate, moderator analysis will also be performed between factors, which may influence the outcome, including, but not limited to, study methodology, study size, year of publication, cohort type, relevant methodology (such as radical prostatectomy vs biopsy studies) or visibility definition. Analytics will be performed as outlined by Wang.[18] In the event that there is insufficient data to conduct a meta-analysis, only thematic synthesis will be performed.

### Patient and public involvement

No patients were involved in this study.

## DISCUSSION

Over the past decade, mpMRI has been widely adopted as the risk stratification tool of choice for men at risk of prostate cancer. This has become particularly true following incorporation of prebiopsy mpMRI into national and international prostate cancer guidelines.[19 20] As such, it is now crucial that we better appreciate the characteristics of prostate cancers that are detected and missed by mpMRI.[4] Through systematic review, we aim to identify commonality between pathology-based studies which have investigated mpMRI-visible and mpMRI-invisible tumours. Our results will enhance the understanding of the histopathological features that influence prostate cancer visibility on mpMRI, particularly beyond increased Gleason grade and high-tumour volume.

Estimates for proportions of significant mpMRI-undetected prostate cancer vary between studies[21]; however, it has become clear that microscopic tumour composition is likely an important factor in determining mpMRI signal (and therefore disease detection). Our planned thematic synthesis will assimilate the evidence surrounding the microscopic basis of tumour conspicuity, including the following considerations. Diffusion

of water is a key principle of MRI signal generation; this is affected by alterations in tissue microenvironment and cellular structure, which may hamper free movement of water molecules.[22] This concept theoretically explains the disparate MRI appearances of the different zones of the prostate (including the peripheral zone and normal central gland) and is also likely to be crucial in the mpMRI visibility of prostate cancer.[23] Indeed, an increased architectural density has been observed in mpMRI-visible tumours, with an increased proportion of cancer cells and decreased proportions of stroma and luminal spaces.[24] This may, in turn, explain visibility of these tumours on mpMRI, through increased restriction of water diffusion within dense tumour tissue (thus generating high signal on the diffusion weighted mpMRI sequence). Variations in tumour density may explain variable ability of mpMRI to estimate gross tumour volume and may explain why alternate imaging modalities, such as prostate specific membrane antigen positron emission tomography/computed tomography (PSMA-PET/CT), may provide more accurate representation of volume by targeting tumour-specific markers.[25 26] It also appears that mpMRI-visible tumours have raised microvessel density,[27] which is cohesive with genetic studies that also have demonstrated an enrichment of vascular endothelial growth factor in these tumours.[28 29] An increased level of vasculature could potentially explain tumour visibility on mpMRI, through increased concentrations of mpMRI-contrast agent (gadolinium) in tumour vessels, thus generating higher MRI signal on the dynamic contrast sequence. Finally, there is mixed evidence regarding the mpMRI visibility or the mpMRI invisibility of particular prostate cancer subtypes, including cribriform pattern[30 31] and ductal cancer.[32] This is potentially concerning, as these cancer subtypes are associated with increased disease aggression and poor clinical outcome. Our planned systematic review (described in this protocol) will highlight and discuss these postulated features, among many others, in detail.

Another major aim of this review was to catalogue heterogeneity within this literature. We have recently reviewed the genetic landscape that underpins the conspicuity of prostate cancer on mpMRI and found a high degree of diversity among study methodologies, radiological scoring systems used and definitions of visibility and clinical significance.[33] This heterogeneity is further complicated by the substantial differences in scan quality observed between different centres.[34] In this proposed review, we will systematically identify sources of heterogeneity, which we hope will guide future studies in this sphere.

In summary, this systematic review will combine the extant evidence in this emerging field, for the first time. Collation and analysis of these data will enrich our understanding of the additional histopathological factors (beyond tumour grade and size) that contribute to the conspicuity of prostate cancer on mpMRI. Additionally, this process will also help reveal the potential clinical role that these factors play in both diagnosis and treatment (eg, during planning focal therapy and radiotherapy)[35 36] and will aid identification of important avenues for future research.

## Trial status

- ► Preliminary searches: started.
- ► Piloting of the study selection process: started.
- ► Formal screening: started.
- ► Data extraction: not started.
- ► Risk of bias assessment: not started.
- ► Data analysis: not started.

## Draft of search strategy for MEDLINE, Embase, PubMed and Cochrane databases

(((((prostat* NOT prostatitis) AND ("cancer" OR tumo?r* OR malignancy*)) AND ("MRI" OR "MRI" OR "multiparametric MRI" OR "mpMRI" OR "mp-MRI")) AND (patholog* OR histopatholog* OR histo* OR "IHC" OR "Immunohistochemistry")) AND visib*).ti,ab

## ETHICS AND DISSEMINATION

Due to the nature of the study, there are no relevant ethical concerns and informed consent will not be required. The protocol and systematic review will be disseminated via a peer-reviewed journal.

**Author affiliations**
¹UCL Division of Surgery and Interventional Science, University College London, London, UK
²Department of Pathology, University College London Hospitals NHS Foundation Trust, London, UK
³School of Healthcare Sciences, College of Biomedical and Life Sciences, Cardiff University, Cardiff, UK
⁴Department of Radiology, University College London Hospitals NHS Foundation Trust, London, UK

**Contributors** The authors' contribution includes, but is not limited to, the following: JMN, LMCE and BSSS drafted the manuscript and created the study concept. RB, AK, AF, DK, HCW and ME provided supervision and guidance during the study. All authors reviewed and approved the manuscript in its current form.

**Funding** JMN was funded by the Medical Research Council (MR/S00680X/1). LMCE received funding from Prostate Cancer UK. BSS received funding from the Rosetrees Trust.

**Competing interests** JMN received funding from the MRC. BSSS received funding from the Rosetrees Trust. LMCE received funding from PCUK. HCW received funding from PCUK, the Urology Foundation and the Rosetrees Trust. AK, AF and ME have stock interest in Nuada Medical. ME received funding from NIHR-i4i, MRC, Sonacare, Trod Medical, Cancer Vaccine Institute and Sophiris Biocorp for trials in prostate cancer. ME is a medical consultant to Sonacare, Sophiris Biocorp, Steba Biotech, GSK, Exact Imaging and Profound Medical. Travel allowance was previously provided from Sanofi Aventis, Astellas, GSK and Sonacare. ME is a proctor for HIFU with Sonacare Inc. and paid for training other surgeons in this procedure.

**Patient and public involvement** Patients and/or the public were not involved in the design, conduct, reporting or dissemination plans of this research.

**Patient consent for publication** Not required.

**Provenance and peer review** Not commissioned; externally peer reviewed.

and indication of whether changes were made. See: https://creativecommons.org/licenses/by/4.0/.

ORCID iDs

Joseph M Norris http://orcid.org/0000-0003-2294-0303
Benjamin S Simpson http://orcid.org/0000-0003-3685-6110
Daniel Kelly http://orcid.org/0000-0002-1847-0655

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
