## [Reviewer comments · BMJ Open]

ARTICLE DETAILS

TITLE (PROVISIONAL)	Histopathological features of prostate cancer conspicuity on multiparametric MRI: a protocol for a systematic review and meta-analysis
AUTHORS	Norris, Joseph; Carmona Echeverria, Lina; Simpson, Benjamin Scott; Ball, Rhys; Freeman, Alex; Kelly, Daniel; Kirkham, Alex; Whitaker, Hayley; Emberton, Mark

VERSION 1 – REVIEW

REVIEWER	Anca L Grosu Department of Radiation Oncology University Medical Center Freiburg, Germany
REVIEW RETURNED	16-Jul-2020

GENERAL COMMENTS	The manuscript "Histopathological features of prostate cancer conspicuity on multiparametric MRI: a protocol for a systematic review and meta-analysis" explains in detail how Norris et al. plan to conduct a review about histopathological features of MRI-visibility. The issue of clinically significant but mpMRI-invisible Prostate cancer is of relevance and the manuscript is well written. Furthermore, the described methodology is of high quality and is likely to create a valuable review about this filed of research. However, some aspects need to undergo further explanation or consideration: 1. The abstract mentions the suspected important clinical implications in diagnosis and prognosis. However, this aspect is poorly elaborated throughout the manuscript and should be revised.2. The review is focused on the imaging method mpMRI, nevertheless progress in PSMA-PET/CT availability and evidence about PSMA-PET/CT performance should be shortly mentioned in order to assess the expected clinical benefit of the review results. Moreover, the comparison between mpMRI and PSMA PET and slide by slide histology should be discussed (s. Bettermann et al, Zamboglou et al).3. Similarly, improvements in non-invasive PCa characterization e.g. radiomics and their impact on PCa diagnostic and therapy should be considered.4. Page 4, line 3/4 "...included studies will be undergo analysis..." should be grammatically revised.
---

REVIEWER	Matthijs J Scheltema Department of Urology, Amsterdam UMC, Amsterdam, the Netherlands
-----------------	--

	St Vincent's Prostate Cancer Centre, Sydney, Australia
REVIEW RETURNED	27-Jul-2020

GENERAL COMMENTS	The authors are addressing an import clinical dilemma; what are histopathological characteristics of prostate cancer that is being missed on MRI. This may lead to improvement of existing imaging protocols and subsequently prostate cancer diagnostics. Therefore I would like to encourage the performance of the proposed systematic review. Below are my concern regarding the existing protocol. TITLE None ABSTRACT systematic review registration number is missing INTRODUCTION No comments METHODS The inclusion criteria for studies is in my opinion not specific enough. Within the existing literature regarding prostate MRI and histological outcomes there is a variety of outcomes. The definitions of significant prostate cancer is different between trial, and this implicates what these studies classify as MRI visible or non-visible. To me its not clear how the authors are dealing with this with the inclusion of studies. In line, there are studies using whole-mount prostatectomy specimen as verification versus systematic versus template biopsies. How are the authors correcting for the differences in outcomes for that? Moreover, there is a huge difference in the quality of mpMRI between centers. The authors of this protocol work in one of the most renown center for prostate imaging. The results that expert centers get with mpMRI does not always reflect the outcomes in community-acquired mpMRI's. In the protocol the authors should provide guidance in how they want to correct for the inherent bias that the above heterogeneity will cause. I'm hesitant that the current literature will allow a meta-analysis. In my humble opinion a systematic review only will potentially be a better suitable design. The Newcatle-Ottawa score will probably not fully include the bias that these differences in trial design / definitions will cause. RESULTS N/A DISCUSSION
---

	Discussion should include the comments I've discussed above (see methods). FUNDING No comments
--	--

VERSION 1 – AUTHOR RESPONSE

Reviewer 1:

The manuscript “Histopathological features of prostate cancer conspicuity on multiparametric MRI: a protocol for a systematic review and meta-analysis” explains in detail how Norris et al. plan to conduct a review about histopathological features of MRI-visibility. The issue of clinically significant but mpMRI-invisible prostate cancer is of relevance and the manuscript is well written.

Furthermore, the described methodology is of high quality and is likely to create a valuable review about this field of research.

Thank you for the kind comments on the relevance and importance of our systematic review, as well as our methodological rigour and writing style.

However, some aspects need to undergo further explanation or consideration:

1. The abstract mentions the suspected important clinical implications in diagnosis and prognosis. However, this aspect is poorly elaborated throughout the manuscript and should be revised.

Where appropriate, throughout the manuscript, we have now added additional description elaborating on the clinical relevance of the MRI visibility of prostate cancer (and the factors that influence it).

2. The review is focused on the imaging method mpMRI, nevertheless progress in PSMA-PET/CT availability and evidence about PSMA-PET/CT performance should be shortly mentioned in order to assess the expected clinical benefit of the review results. Moreover, the comparison between mpMRI and PSMA PET and slide by slide histology should be discussed (s. Bettermann et al, Zamboglou et al).

Thank you for highlighting the progress made with PSMA-PET/CT. We have now commented on the relevance of PSMA-PET/CT in this field. We have made specific reference to how MRI may underestimate tumour volume compared to PSMA/PET, reflecting potential density changes at the tumour boundary. We have also referenced the papers suggested by the reviewer.

3. Similarly, improvements in non-invasive PCa characterization e.g. radiomics and their impact on PCa diagnostic and therapy should be considered.

As suggested, we have now highlighted the use of other non-invasive modalities in the diagnosis and treatment planning for suspected prostate cancer.

4. Page 4, line 3/4 "...included studies will be undergo analysis..." should be grammatically revised.

Thank you for highlighting this – we have now revised this line.

Reviewer 2:

The authors are addressing an import clinical dilemma; what are histopathological characteristics of prostate cancer that is being missed on MRI. This may lead to improvement of existing imaging protocols and subsequently prostate cancer diagnostics. Therefore, I would like to encourage the performance of the proposed systematic review.

Thank you very much for the kind comments regarding the importance of our proposed systematic review and the potential impact that this may have on clinical practice.

Below are my concerns regarding the existing protocol.

Thank you for raising your concerns – we have addressed these sequentially below.

TITLE:

None.

Thank you.

ABSTRACT:

Systematic review registration number is missing.

We have now added the systematic review registration (PROSPERO) number to the abstract (CRD42020176049).

INTRODUCTION:

No comments.

Thank you.

METHODS:

The inclusion criteria for studies is, in my opinion, not specific enough. Within the existing literature regarding prostate MRI and histological outcomes, there is a variety of outcomes. The definitions of significant prostate cancer are different between trials, and this implicates what these studies classify as MRI visible or non-visible. To me it is not clear how the authors are dealing with this with the inclusion of studies. In line, there are studies using whole-mount prostatectomy specimen as verification versus systematic versus template biopsies. How are the authors correcting for the differences in outcomes for that? Moreover, there is a huge difference in the quality of mpMRI between centers. The authors of this protocol work in one of the most renowned centers for prostate imaging. The results that expert centers get with mpMRI does not always reflect the outcomes in community-acquired mpMRIs. In the protocol, the authors should provide guidance in how they want to correct for the inherent bias that the above heterogeneity will cause. I'm hesitant that the current literature will allow a meta-analysis. In my humble opinion, a systematic review only will potentially be a better suitable design.

Thank you to the reviewer for providing such detailed, thorough feedback. A number of valuable points have been raised and we believe that we have addressed these concerns below. Overall, the reviewer's concerns are focussed on heterogeneity of data (a problem faced in all systematic reviews), particularly with regards to: definitions of clinical significance, approach to pathological sampling, and quality of mpMRI. In our planned systematic review, we will note and acknowledge all identifiable sources of bias and heterogeneity, and then factor these in when drawing conclusions. However, there are additional considerations (see below).

We fully expect the methodologies of included studies will vary (for instance, in terms of definitions of tumour visibility, mpMRI scoring schemes, mpMRI magnet strengths & acquisition protocols, definitions of clinical significance, pathological reference standards, and, of course, methods of quantifying a particular histopathological feature), as will the quality of mpMRI from various centres. However, these difference between studies are unavoidable, and importantly, do not preclude compilation or comparison of derived data, as the broad over-arching principles (e.g. "MRI-visible clinically significant cancer") will be sustained, regardless of smaller methodological differences. The key feature of our review is that we will carefully document, account and discuss methodological differences, and factor these into our discussion and conclusion.

Indeed, our recording of methodological heterogeneity across the included studies will be a useful aspect of this review, allowing us to make recommendations for future studies, to adopt certain practices to maximise data comparability. This approach was useful in a similar review that we have recently published on the genetic characteristics of MRI-visible and MRI-invisible prostate cancer (doi.org/10.1016/j.euros.2020.06.006). Additionally, we will use mathematical methods to assess impact from potential confounders. Moderator analysis will help us to determine whether individual study methodology will significantly impact estimated frequencies. Additionally, we will also perform a leave-one-out (LOO) analysis removing each

study sequentially, each time measuring model heterogeneity through metrics such as: externally studentised residuals, difference in fits values (DFFITS), Cook's distances, covariance ratios, LOO estimates of the amount of heterogeneity, LOO values of the test statistics for heterogeneity, hat values and weights. This robust analysis will help us to reveal whether a study with a particular methodology, low inter-observer agreement, or reduced quality of MRI (e.g. in inexperience centres) may represent an outlier and as such, should be removed from the analysis.

Lastly, if statistical meta-analysis is not possible with available data, then we will simply conduct a systematic review (with thematic synthesis), as the reviewer has suggested.

We have now referred to the above points and discussion in our revised manuscript.

The Newcastle-Ottawa score will probably not fully include the bias that these differences in trial design / definitions will cause.

We agree that risk of bias tools (including, the Newcastle-Ottawa score) may not highlight all possible sources of bias. However, we have previously used this tool effectively in a similar systematic review (doi.org/10.1016/j.euro.2020.06.006) with acceptable results, assessing similar levels of bias. Furthermore, we hope that the above methodological steps in our meta-analysis will also highlight and account for some of the additional biases present in these studies.

RESULTS:

N/A.

Thank you.

DISCUSSION:

Discussion should include the comments I've discussed above (see methods).

We appreciate the reviewer's useful suggestions. In addition to extra detail in the methods section, we have now discussed these points within the discussion.

FUNDING:

No comments.

Thank you.

VERSION 2 – REVIEW

REVIEWER	Anca Grosu Department of Radiation Oncology Medical Center - University of Freiburg
REVIEW RETURNED	07-Sep-2020

GENERAL COMMENTS	The authors performed the requested changes and better worked out the clinical implications of the review's topic. Only the aspect of non-invasive tissue characterization is still missing. However the described search methodology will likely lead to inclusion of articles regarding this topic.
---

REVIEWER	Matthijs JV Scheltema Department of urology, Amsterdam UMC, Amsterdam, the Netherlands St. Vincent's Prostate cancer centre, Sydney, Australia Garvan institute of medical sciences, Sydney, Australia
REVIEW RETURNED	26-Aug-2020

GENERAL COMMENTS	I would like to thank the authors for their revision of the manuscript. I'm satisfied with their reply and would like to use this opportunity to wish them good fortune in the conduct of their design research project.
--